# Development of CO$_2$-Sensitive Viscoelastic Fracturing Fluid for Low Permeability Reservoirs: A Review

Allah Bakhsh [1,2], Liang Zhang [1,3,*], Huchao Wei [1], Azizullah Shaikh [2], Nasir Khan [2], Zeeshan Khan [2] and Ren Shaoran [1,*]

1 School of Petroleum Engineering, China University of Petroleum (East China), Qingdao 266555, China; lb1702016@s.upc.edu.cn (A.B.); s19020114@s.upc.edu.cn (H.W.)
2 Balochistan University of Information Technology, Engineering and Management Sciences, Quetta 87300, Pakistan; hafiz.azizullah@buitms.edu.pk (A.S.); nasir.khan1@buitms.edu.pk (N.K.); zeeshan.khan@buitms.edu.pk (Z.K.)
3 Key Laboratory of Unconventional Oil & Gas Development, Ministry of Education, China University of Petroleum (East China), Qingdao 266555, China
* Correspondence: zhlupc@upc.edu.cn (L.Z.); rensr@upc.edu.cn (R.S.)

**Abstract:** There are economic and technical challenges to overcome when increasing resource recovery from low permeability reservoirs. For such reservoirs, the hydraulic fracturing plan with the development of clean and less expensive fracturing fluid plays a vital aspect in meeting the energy supply chain. Numerous recent published studies have indicated that research on worm-like micelles (WLMs) based on viscoelastic surfactant (VES) fluid has progressed substantially. This study looks at the development of CO$_2$-sensitive viscoelastic fracturing fluid (CO$_2$-SVFF), its applications, benefits, limitations, and drawbacks of conventional fracturing fluids. The switchable viscoelasticity of CO$_2$-SVFF system signifies how reusing of this fluid is attained. Compared to conventional surfactants, the CO$_2$-SVFF system can be switched to high viscosity (to fracture formation and transporting proppants) and low viscosity (easy removal after causing fracture). The effect of pH, conductivity, temperature, and rheological behaviors of CO$_2$-SVFFs are also highlighted. Further, the aid of Gemini surfactants and nanoparticles (NPs) with low concentrations in CO$_2$-SVFF can improve viscoelasticity and extended stability to withstand high shear rates and temperatures during the fracturing process. These studies provide insight into future knowledge that might lead to a more environmentally friendly and successful CO$_2$-SVFFs in low-permeability reservoirs. Despite the increased application of CO$_2$-SVFFs, there are still several challenges (i.e., formation with high-temperature range, pressure, and salinity).

**Keywords:** low permeability reservoirs; fracturing fluid; worm-like micelles; rheology; CO$_2$-sensitive viscoelastic surfactant

## 1. Introduction

Hydraulic fracturing is an efficient stimulation method for generating highly conductive conduits between wellbores and low permeability reservoirs [1,2]. Pressurized fluid containing proppants is injected into the low permeability reservoirs to keep the fracture open [3,4]. This method includes constructing and enlarging conductive conduits through which the hydrocarbon can flow easily. The fracture network formed improves the reservoir rock's hydraulic conductivity while increasing the surface area available for hydrocarbon production [5,6]. Fracturing fluids are used to create artificial fractures in reservoirs and transport proppant particles into the cracks to improve formation conductivity [7–9]. Pad or prepad fluids are initially pumped into the formation to create the fracture geometry. After the fracture geometry has been generated, a more fluid-carrying proppant is transported into the fractures. The proppant prevents fracture closure by providing a conductive conduit for hydrocarbons to flow back into the wellbore [10–12]. Proppant transport is

influenced by the fracturing fluid's rheology, proppant properties, and fracture geometry. The fracturing fluid must be economical, compatible with the formation, residue-free, have a lot of fracturing experience and withstand high temperatures and shear rate [13].

In the past, oil-based fracturing fluids were utilized in hydraulic fracturing. Since oil-based fracturing fluids have been linked to environmental and safety problems, the industry has been developing more environmentally acceptable water-based fracturing fluids [14]. Several polymers were utilized to improve the viscosity of the fracturing fluid due to low water viscosity. Borate, titanium, and zirconium cross-linkers increase polymers' gel strength and viscosity [15–17]. On the other hand, the polymer fracturing fluid leaves insoluble residues that reduce the formation's permeability [18–20]. Limited sand carrying capacity is the other drawback of this fracturing fluid [21]. Chemical EOR [22], gravel packing [23], drilling [24,25], and hydraulic fracturing [7] can all benefit from the use of viscoelastic fluids that are polymer-free. Shell developed viscoelastic surfactants (VESs) to solve the polymer fluid problems in hydraulic fracturing [26,27]. VES fracturing fluids provide several advantages over polymer fracturing fluids [28,29]. The advantages including the absence of insoluble residues, low-pressure friction, gel breaking capacity, ease of preparation, good proppants carrying capacity, and minimal formation damage [30,31]. At high shear rates and temperatures, most VES fracturing fluids, on the other hand, are less durable [31].

This review study is divided into five sections. Surfactants with Viscoelastic Properties, $CO_2$-SVFFs, Application of $CO_2$-SVFFs, Properties of $CO_2$-SVFFs, and their benefits and limits are discussed.

### 1.1. Surfactants with Viscoelastic Properties

Smaller than guars, viscoelastic surfactants (VESs) have less than a thousand molecular weights. VES-based fluids are polymer-free fluids widely used in hydraulic fracturing [13,32]. Under certain conditions, VESs can self-assemble into colloidal forms recognized as worm-like micelles due to repelling and attractive interactions between the surfactants and the solvents. Worm-like micelles with contour lengths varying from a few nanometers to several micrometers [33,34]. The worm-like micelles tangle into a transient network beyond a threshold concentration c*, identical to a solution of flexible polymers with exceptional viscoelastic properties [35,36]. However, in contrast with polymers, it breaks and recombines during the dynamic process.

The hydrophobic tails of surfactant molecules orient themselves toward the inner of the micelles and away from the polar medium (water), resulting in the surfactant molecule's aggregation [37]. Worm-like micelles can provide fracturing fluid with no residues, high conductive stimulation, and high-quality viscoelastic proppant transport characteristics [19,38]. Various self-assembly theories govern micellar assemblages and VES solutions with appropriate interactions and additives under multiple circumstances [39–41]. Critical packing parameters can identify the micelle arrangement in a solution phase. $P = v/a_ol$, where $v$ is the volume of the surfactant's hydrophobic tail, $l$ is the length of the tail, and $a_o$ is the surfactant head group's optimal surface [42]. Different micelle configurations in the bulk solution as shown in Figure 1.

The aggregate geometry is stronger when the high packing value [43,44]. The chemical composition of a certain surfactant is often used to develop VES fluids, for instance, anionic, [45,46], cationic [20,47], nonionic [48,49], Zwitterionic/amphoteric [50,51] or a combination of surfactants; cationic/anionic [31,52], nonionic/anionic [53] and Zwitterionic/anionic [54,55]. The sandstone reservoir is better wet by anionic surfactants than cationic surfactants and is less expensive and easier to biodegrade. On the other hand, anionic surfactants exhibit temperature instability. Zwitterionic surfactants have the combined presence of expensive but thermal stability anionic and cationic centers in their heads than other surfactants [44,54]. Zwitterionic surfactants use two different surfactants to increase VES fluid performance [55,56]. Effect of various micelles on surfactant activity and surfactant rheological behavior based on different parameters, for example, surfactant

concentration [57], concentration and type of salt [58–60], alkali [60], acid [61], $CO_2$ [32,62], pH [63,64], temperature [60,65], occurrence of redox reactions [66,67], and light irradiation [68,69]. Some additives may even be required to support surfactants assembled into distinct molecular structures to improve viscoelasticity spontaneously [70,71]. Adjusting the appropriate additive-VES solution level is required for efficient viscoelastic behavior (Figure 1). The morphology of worm-like micelles (WLMs) is influenced by different degrees of ionic strength by salt concentration and types.

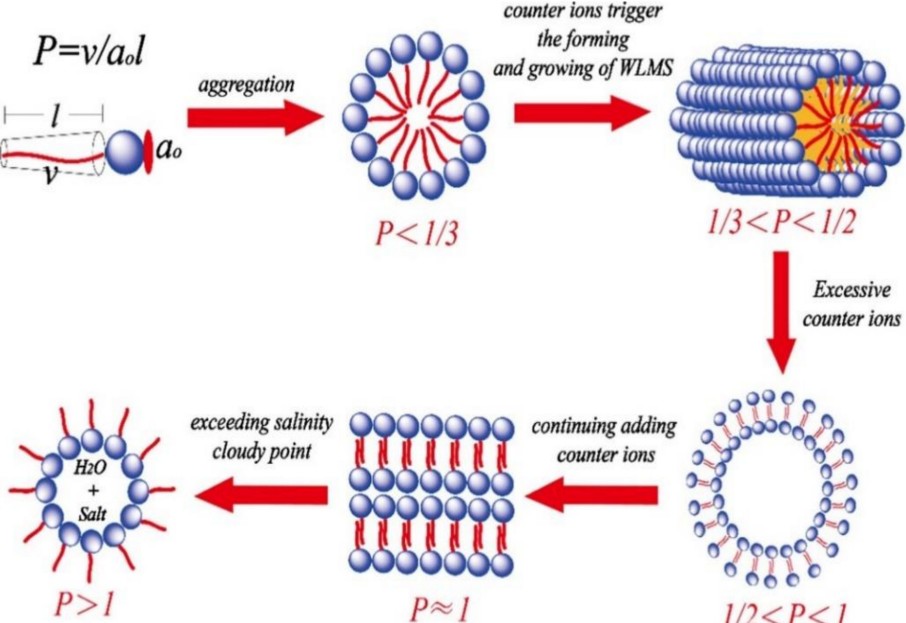

**Figure 1.** The effect of additional counter ions on the behavior of the VES phase [42].

The degree of contra-ion binding is determined by the hydrophobicity and micellization of the counterion. Electrostatic screening for the repulsion between the surfactant head groups occurs by oppositely charged counterion ions due to organic or inorganic salts mixed with various surfactants. Halide ions from organic salts such as $Cl^-$ or $Br^-$ are considered to be adsorption to the micelle surface based on weak to relatively weak interactions with the surfactant cations. Whereas inorganic salts such as sodium salicylate (NaSal) are tightly bound to the surface head groups, micelle form transition improves when salt concentration decreases [72,73]. The micelle disrupts under certain shear forces, yet it assembles its initial form again after applying shear stress to eliminate [74], in contrast to the polymer degradation process, which can reversibly deteriorate after shear stress [75,76]. Hydrocarbons or formation water can break the structure of the VES fluid, reduce its viscosity, and thus, no chemical or chemical breaker requirement [77,78]. VES-based fracturing fluids are more stable over a certain salt concentration, and the gel can be broken easily by diluting with the formation fluids. Hydrophobic substances such as oil or gas are dissolved in the hydrocarbon core of the micelle, and the structure stretches and fractures into smaller spherical micelles [30]. The VES fluid does not break completely in dry gas or low oil saturation reservoirs; hence an internal breaker is necessary. An internal breaker distracts the VES micellular structure and reduces viscosity [70,79]. In addition, low molecular weight alcohols or oxidizing breakers may disintegrate WLMs into non-viscous spherical micelle or destroy the structure of surfactant molecules [30]. Numerous researchers used various improvements in the study of the quality of VES fracturing fluid (VFF) [4,80]. In addition, experimentation and field applications are less documented [79,81]. The WLMs in micellar fluids are remarkably similar to polymers; hence they are constantly dispersed and recombined at equilibrium [82]. However, the high cost, environmental problems, high temperatures, and high salt concentration are

some of the limitations of mostly conventional VES fluids [83,84]. Viscoelastic surfactant system is always resistant to poor temperature. At high temperatures, the molecular thermal motion rates increase, molecules' strength decreases, and the CMC of surfactants rises geometrically [85]. The primary disadvantages of VES fracturing fluids, on the other hand, are their high costs, limited temperature resistance, and potential environmental concerns [32,86].These issues must be overcome by re-usable viscoelastic surfactant (rVES) fluid, VES/$CO_2$ fluid, VES/NP fluid, and Gemini-VES fluid [87–89]. In Gemini surfactants, a spacer group connects 2 or 3 hydrophilic head and hydrophobic tail groups [90].

Several review studies on various viscoelastic surfactants used for hydraulic fracturing procedures have been published. The study of smart worm-like micelles (SWLMs), which can reversibly change the rheological behavior from low viscosity to higher viscoelasticity, has recently gotten much attention [91–94]. Temperature, pH, light, and redox potential have all been used previously to "turn" micellar assemblies ON and OFF. Each has its disadvantages, such as spatial restriction, excessive usage of energy, or contamination, limiting the process's reversibility [88]. Conventional triggers have drawbacks; for instance, surfactant space on thermos-responsive triggers would be restricted [95].In the field of fracturing fluids, VES is a study issue; nevertheless, fracturing fluid flow-back can create significant environmental concerns [96].

### 1.2. $CO_2$-Sensitive Viscoelastic Surfactants

$CO_2$–SVFF solution offers better benefits because environmentally friendly and plentiful $CO_2$ may be used as a green trigger in practical applications. $CO_2$ has many other advantages that may be used in hydraulic fracturing fluid, including being inexpensive, non-hazardous, energy-efficient, and easy to remove from the system [31]. The current study will explain how $CO_2$ induced in viscoelastic surfactants fluids improves its properties compared to previous research.

Mathew Samuel et al. [97] worked on the development in fracturing fluids, introducing a polymer-free VES fluid that is $CO_2$-compatible. R. Hall et al. [37] developed a novel multi-surfactant clean fracturing fluid system to provide a strong surfactant-based fracturing fluid with the addition of $CO_2$. The field test findings of his study showed that the fluid system is $CO_2$ compatible and exhibits all of the attributes generally associated with VES systems. Adjusting the appropriate additive-VES solution concentration is required to properly manage the viscoelastic behavior as shown in Figure 2. According to multiple research publications on the subject, $CO_2$ is useful in extracting oil from low permeability oil reservoirs because it increases oil swelling, decreases viscosity, and vaporizes components of crude oil as it is carried through porous rock [98].

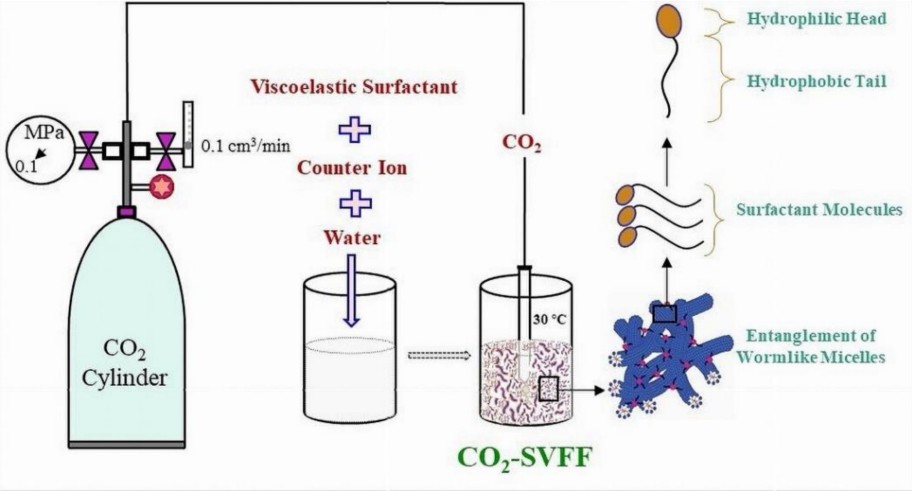

**Figure 2.** Sketch diagram displaying the preparation process of $CO_2$-SVFF.

CO$_2$-SVFFs have attracted many researchers' attention in the last few years [99]. CO$_2$ can improve the viscoelasticity and aggregate structure of VES fluids. CO$_2$-SVFFs containing WLMs are extraordinarily viscous and elastic because of their physical association and entangled structures [41]. In four different forms, CO$_2$-responsive functional groups are found in conventional CO$_2$-responsive surfactants (tertiary amine, amidine, guanidine, and imidazoline) [100]. Compared to the others, tertiary amines are a traditional chemical product as a conventional functional group. It responds well to CO$_2$ and has simple production methods. As illustrated in Figure 3 below, there have been several CO$_2$-responsive compounds [101].

**Figure 3.** CO$_2$-responsive functional groups that are commonly used [101].

In a fixed proportion, an anionic surfactant and its counterion are added to generate spherical VESs. After CO$_2$ bubbling and protonation, spherical micelles convert into WLMs. The CO$_2$-induced VES showed a reverse transformation after the pH was changed. This fluid can be reused due to its CO$_2$ responsiveness and switchable viscoelasticity. CO$_2$-responsive switchable surfactants can perform two functions at once, fracturing and proppant transport and recovery following fracturing, by switching between high and low viscosity [32]. Inspired by the pseudo-Gemini idea work, Zhao et al. [102] constructed a CO$_2$-sensitive anionic SWLMs system using sodium oleate (NaOA) and the small chemical counter ion 2,6,10-trimethyl-2,6,10-triazaundecane (TMTAD). The 3NaOA-TMTAD solution was initially transparent, having low viscosity and spherical micelles (the primary morphology), as shown in Figure 4. This system exhibits significant viscosity, transparency, and aggregates of mostly wormlike micelles after a duration of CO$_2$ introduction. TMTAD and NaOA were mixed in a 1:3 ratio based on electrostatic interactions to generate a pseudo-Gemini surfactant. The combined solution looks turbid and has low viscosity soon after the excess CO$_2$ is introduced, possibly due to the lower solubility of the NaOA molecules. Sodium hydroxide can be introduced to the system to change the agglomeration

forms of the solution. In the same kind of work, Shaikh et al. [103] presented a Novel $CO_2$-induced clean fracturing fluid (SDS-TMTAD-$CO_2$) created using simulated formation water (23,003 mg $L^{-1}$). Rheological investigations revealed that the apparent viscosity of the fracturing fluid system increases to some degree under high salt conditions and has sufficient self-healing properties against high shear tolerance.

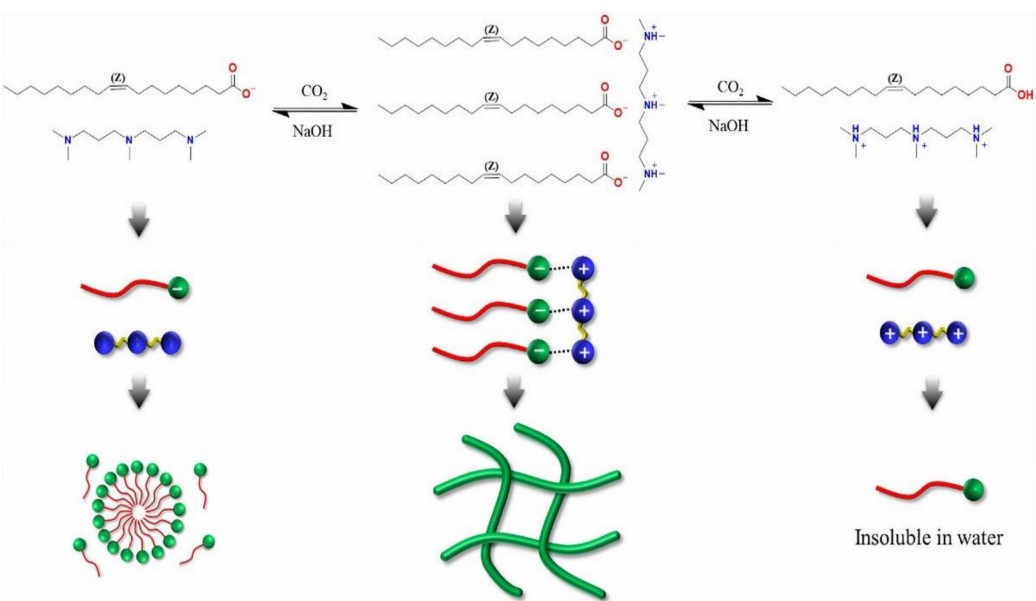

**Figure 4.** The system's self-assembly technique, triggered by $CO_2$, is schematically illustrated [102].

Another same kind of study by Zhang and his co-authors [88] created pseudogemini VES solution of anionic surfactant sodium dodecyl sulfate (SDS) and *N,N,N,N*-tetramethyl-1,3-propanediamine (TMPDA) in the ratio of 2:1. $CO_2/N_2$ altered the rheological behavior between non-Newtonian and Newtonian, as illustrated in Figure 5. The viscosity of a 250 mM SDS-TMPDA aqueous phase is low; however, the fluid viscosity improves after bubbling TMPDA solution with $CO_2$.

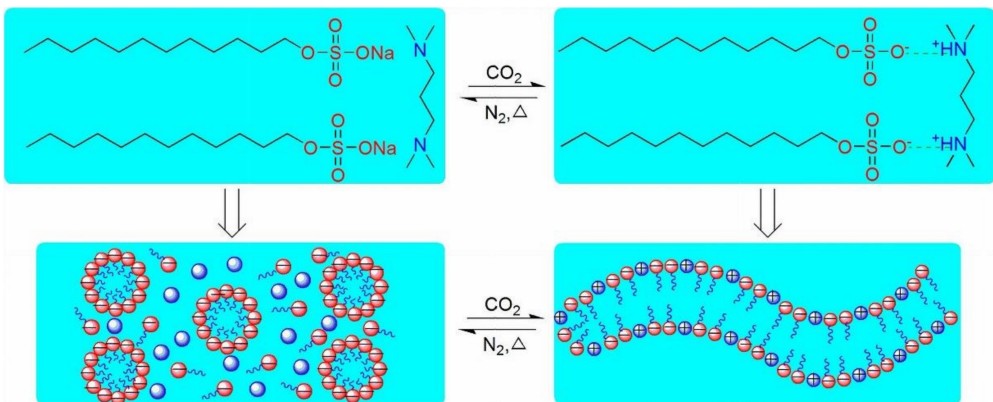

**Figure 5.** The figure depicts the Pseudogemini Surfactant Micellar System's Switching Mechanism [88].

In another research by Zhang et al. [104], the viscoelastic wormlike micelles generated by natural sodium erucate ($C_{22}H_{41}NaO_2$) after bubbling $CO_2$ were studied. In addition, the rheological behaviors of micellar solutions compared to another carbon dioxide–responsive wormlike micelles were investigated. It becomes turbidity when 100 mM erucic acid aqueous solution is combined with bubbling carbon dioxide at 60 °C to saturation (pH = 6.63). After bubbling nitrogen at 85 °C to replace carbon dioxide, the pH raised to 9.22 (9.72 at 100 °C), at which point the solution becomes less viscous and turbidity decreases. After

adding NaOH, a transparent viscoelastic solution was obtained after altering the pH of the solution to 10.63.

A study by Zhang et al. [100] found that the viscoelastic aqueous phase contains only the surfactant *N*-erucamidopropyl-*N*, *N*-dimethylamine (UC22AMPM, Figure 6), and $CO_2$, with no hydrotropes, which are generally required in conventional WLMs to facilitate micelle development via screening strong surfactant binding or electrostatic repulsions between charged surfactant head groups.

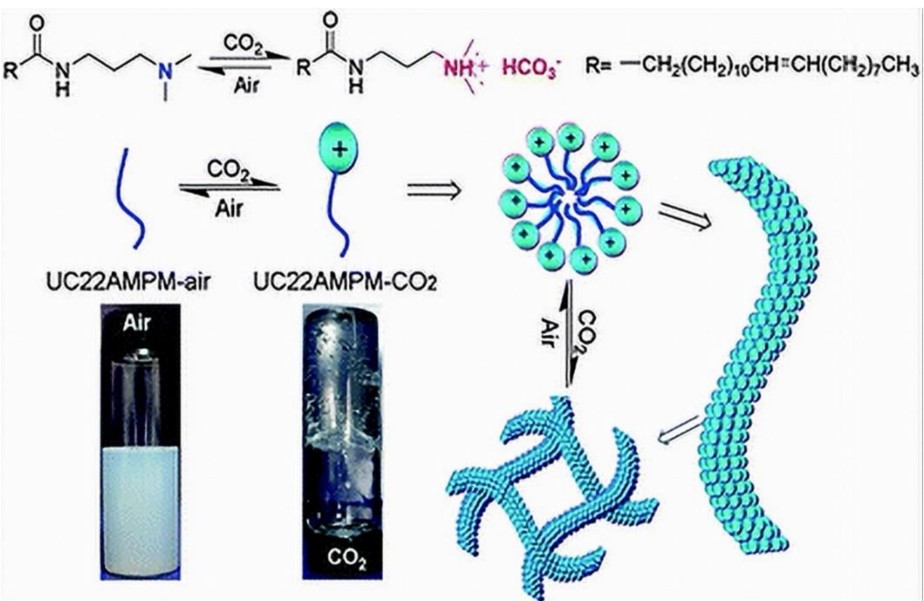

**Figure 6.** The mechanism that governs $CO_2$-air switchable wormlike micelles "Reproduced with permission from Ref. [100]; 2022, Royal Society of Chemistry".

## 2. Application of $CO_2$-Sensitive VES as Fracturing Fluid

There are limitations in slick-water volume fracturing and gas injection to add energy for drainage and displacement [105]. $CO_2$ fracturing as waterless technology can increase reservoir pressure after easy flow back, reduce the solid residue and have low damage characteristics [106]. $CO_2$ diffuses into the matrix, swells the oil, and pushes it into the fractures [37]. $CO_2$ provides many advantages, including reducing formation damage by reducing solid residue and improving production by boosting reservoir pressure following an easy flow back [107].

$CO_2$-SVFFs with no formation damage is very efficient in low permeability reservoirs. $CO_2$-SVFFs can generate fractures in the rock, substantially improving permeability near the wellbore, resulting in a high production rate with low-pressure decline [103]. $CO_2$ dissolves into oil, reducing interfacial tension and oil viscosity while also improving the mobility ratio. The $CO_2$-SVFFs assist in reducing the quantity of water utilized in traditional VES fracturing fluids. $CO_2$ is also non-toxic, non-explosive, and relatively inexpensive [108]. $CO_2$-SVFFs have the potential to improve oil/gas recovery from low-permeability reservoirs significantly.

## 3. Properties $CO_2$-Sensitive Viscoelastic Surfactants

The efficiency of $CO_2$-induced clean fracturing fluid is influenced by pH, conductivity, viscosity, shear rate, salt concentration, pressure, temperature, and rheological behavior.

### 3.1. Effect of pH

As the pH of a solution approach complete ionization, the viscosity of the solution decreases [35,88,109]. The pH level of the solution dropped as $CO_2$ was bubbled into it, and the viscoelastic characteristics altered as the chemical configurations changed [95].

As shown in Figure 7, 150 mM sodium oleate (NaOA) and 50 mM 2,6,10-trimethyl-2,6,10-triazaundecane (TMTAD), a small organic counterion, were introduced in water [102]. At an exact stoichiometric ratio of 3:1, a clear water-like solution with a pH of 12.17 was produced. After $CO_2$ bubbling, a clear, homogeneous, and very dense solution formed with a pH of about 9.17.

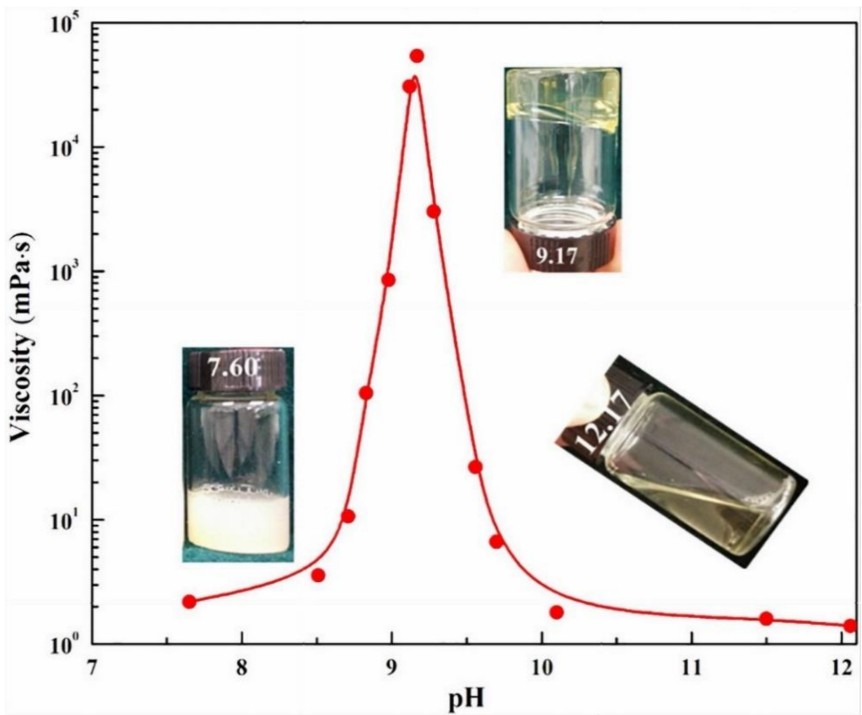

**Figure 7.** At ambient temperature, zero-shear viscosity of 150 mM NaOA and 50 mM TMTAD in aqueous solution with bubbling $CO_2$ (reduction in pH) [102].

In another study, Zhang et al. [110] developed $CO_2$-induced anionic wormlike micellar fluid by combining triethylamine (TEA) with the natural anionic surfactant sodium erucate (NaOEr) at a molar ratio of 3:10 (CNaOEr: CTEA). While $CO_2$ is bubbled into the solution, the pH drops from 12.3 to 10.0. TEA is protonated into a quaternary ammonium salt, promoting micelle formation by reducing electrostatic repulsion between anionic head groups in NaOEr molecules. After $CO_2$ is removed, the quaternized TEA deprotonates back into a non-ionic tertiary amine, resulting in wormlike micelles. As a result, electrostatic repulsion strengthens, converting the viscoelastic fluid to its original low viscosity spherical micellar solution.

### 3.2. Effect of Conductivity

Monitoring the conductivity of a solution while $CO_2$ and then any inert gas bubbled through the solution over different cycles, as shown in Figure 8, can demonstrate the process' reversibility and repeatability [111].

Liu et al. [95] utilized *N*-butyldiethanolamine sodium oleate (BDEA-NaOA) and *N,N*-diethylbutylamine–sodium oleate (DEBA-NaOA) to make two types of $CO_2$-responsive wormlike micelles. The BDEA-NaOA and DEBA-NaOA combination systems generated wormlike micelles after $CO_2$ was bubbled into the solution. As a result, the conductivity and pH level were used to investigate the impact of $CO_2$ on the solutions. After $CO_2$ bubbling into the BDEA-NaOA solution, the conductivity promptly increased and peaked after 20 min, as the tertiary amine protonated to quaternary ammonium salt. At the same time, with the blubbering of $CO_2$, the pH value essentially declined.

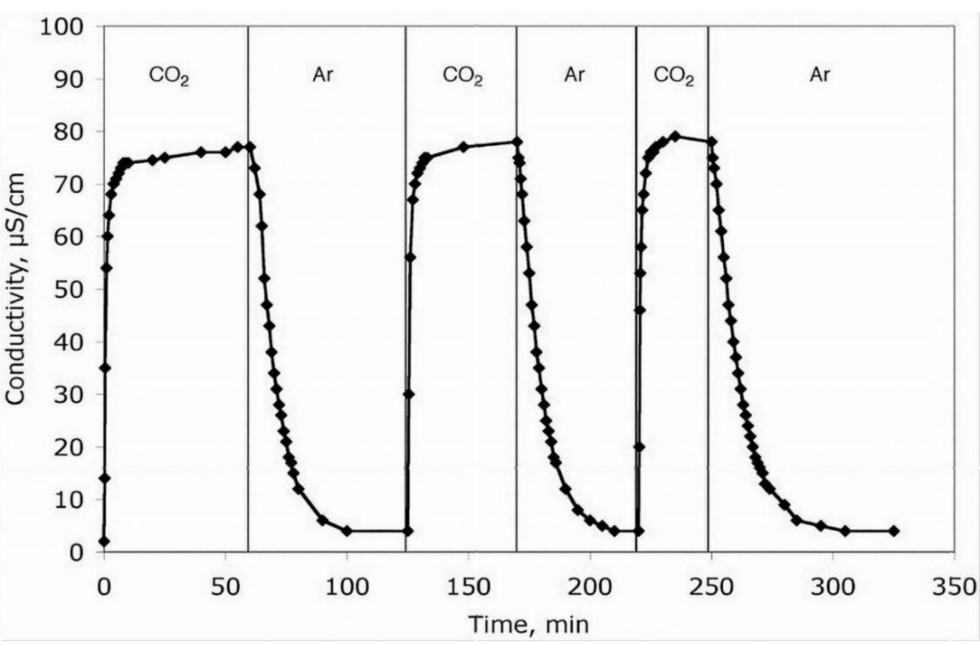

%endadjustwidth

**Figure 8.** The conductivity of a Dimethyl sulfoxide (DMSO) solution as a time-dependent throughout 3 phases of $CO_2$ treatment followed by argon at 23 °C [111].

### 3.3. Effect of Viscosity

The relationship between $CO_2$ and VES fluid viscosity is quite sensitive. VES may turn on and off the high viscosity by adding and removing $CO_2$ from the system. The low-and-high zero-shear viscosity cycle was implemented using two separate ways (bubbling gas and heating) [32], as illustrated in Figure 9. The zero-shear viscosity of a 3% EA solution without $CO_2$ bubbling was 0.0026 Pa·s. Still, after bubbling of $CO_2$, the viscosity jumped to 6 Pa·s. The protonation of VES with $CO_2$ improves the viscosity of $CO_2$-SVFF by increasing the packing parameters by 1/3-1/2. Deprotonation proceeded after $N_2$ bubbling, lowering the viscosity to its original levels.

In other studies, Zhang et al. [112] reported a 2.0 wt% octadecyl dipropylene triamine (ODPTA) dispersion is milky and low-viscosity at room temperature (Figure 10A), but instantly transforms to a clear viscoelastic "gel" following two minutes of $CO_2$ bubbling ("ODPTA-$CO_2$"), which is suitable for trapping bubbles for longer durations (Figure 10B). The "gel" regains its original appearance after replacing $CO_2$ with $N_2$ at 75 °C for approximately 45 min (Figure 10C). Instead of using HCl to bring the pH down to 6.0, a transparent, water-like fluid (Figure 10D) is produced, with no viscoelastic properties.

Su et al. [113], found that aqueous solutions containing both 2-(dimethylamino) ethanol (DMAE) and the surfactant sodium octadecyl sulfate ($C_{18}$SNa) are $CO_2$-responsive (Figure 11a); $CO_2$ tends to cause protonation of DMAE in water, and an enhance in zero-shear viscosity at 60° C (Figure 11b), indicating micelle elongation. The viscosity of VES fluid measurements to verify that the switch is reversible after bubbled $N_2$ by removing the $CO_2$ (Figure 11c). After 50 min, the viscosity reached the same level as distilled water.

### 3.4. Effect of Shear Rate

Resistant to shear is one of the significant parameters of VES fracturing fluids. Its importance arises when fluid with high speed is injected into the formation [114]. The stability of the VES system at high shear is necessary for hydraulic fracturing processes [4]. At high shearing force, the strength of VES fluid should not be weakened. To efficiently withstand shearing force, the pad fluid of the VES fracturing system should regain the imposed high strength before reaching the formation [46]. At high shearing force, the viscoelastic properties of VES fracturing should have the quick ability to recover [20]. A

$CO_2$-SVFF based on the long-tailed surfactant Erucic acid 3-($N$,$N$-dimethylamino) (EADP) was investigated in a study using various ratios of sodium salicylate (NaSal) and maleic acid (MA) [115]. NaSal and "pseudo" Gemini (MA) systems have different types of aggregation of morphologies, such as spherical micelles, worm's preferences, and vesicles.

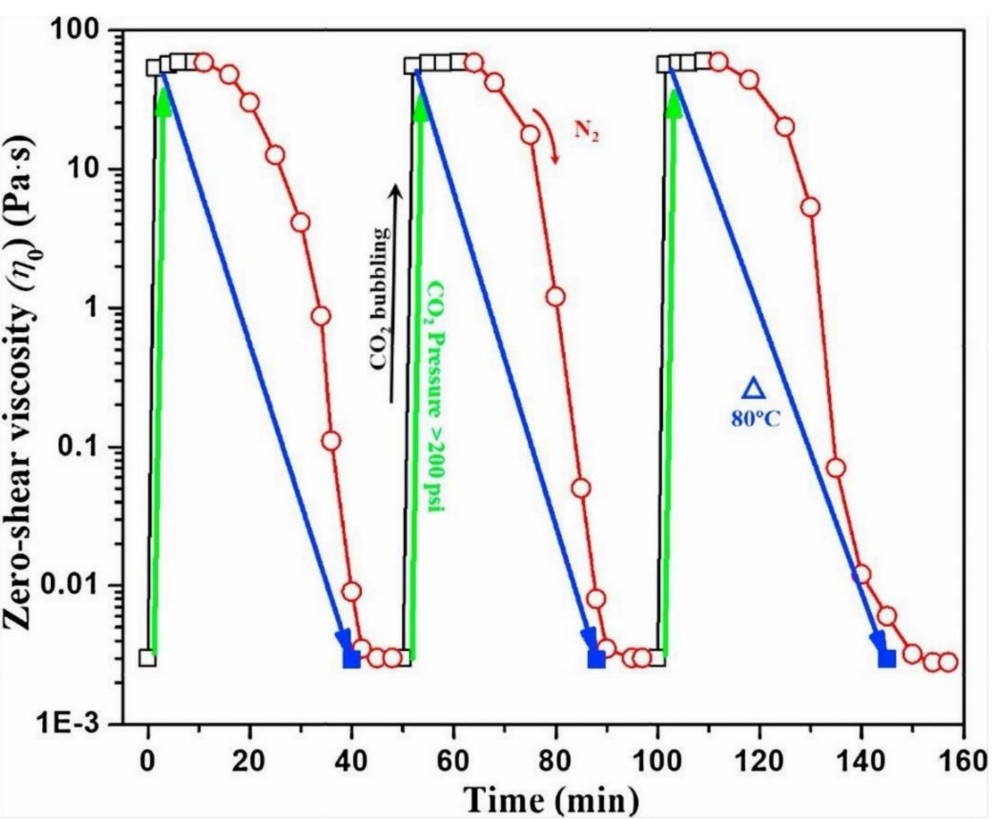

**Figure 9.** At atmospheric temperature and pressure (unless when gel-breaking at 80 °C), 3% erucami-dopropyl dimethylamine (EA) solution with switchable zero-shear viscosity [32].

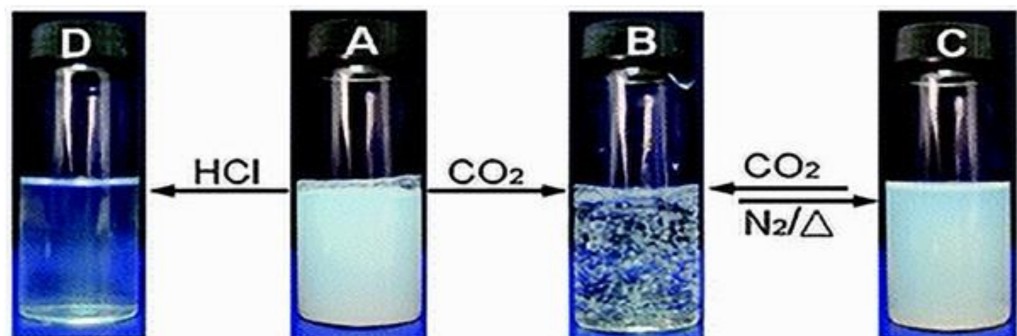

**Figure 10.** 2.0 wt% ODPTA aqueous dispersion: (**A**) the initial dispersion; (**B**) bubbling $CO_2$ (0.1 MPa); (**C**) switching $CO_2$ with $N_2$ (0.1 MPa) at 75 °C; and (**D**) altering the pH with HCl as in (**B**) "Reproduced with permission from Ref. [112]; 2022, Royal Society of Chemistry".

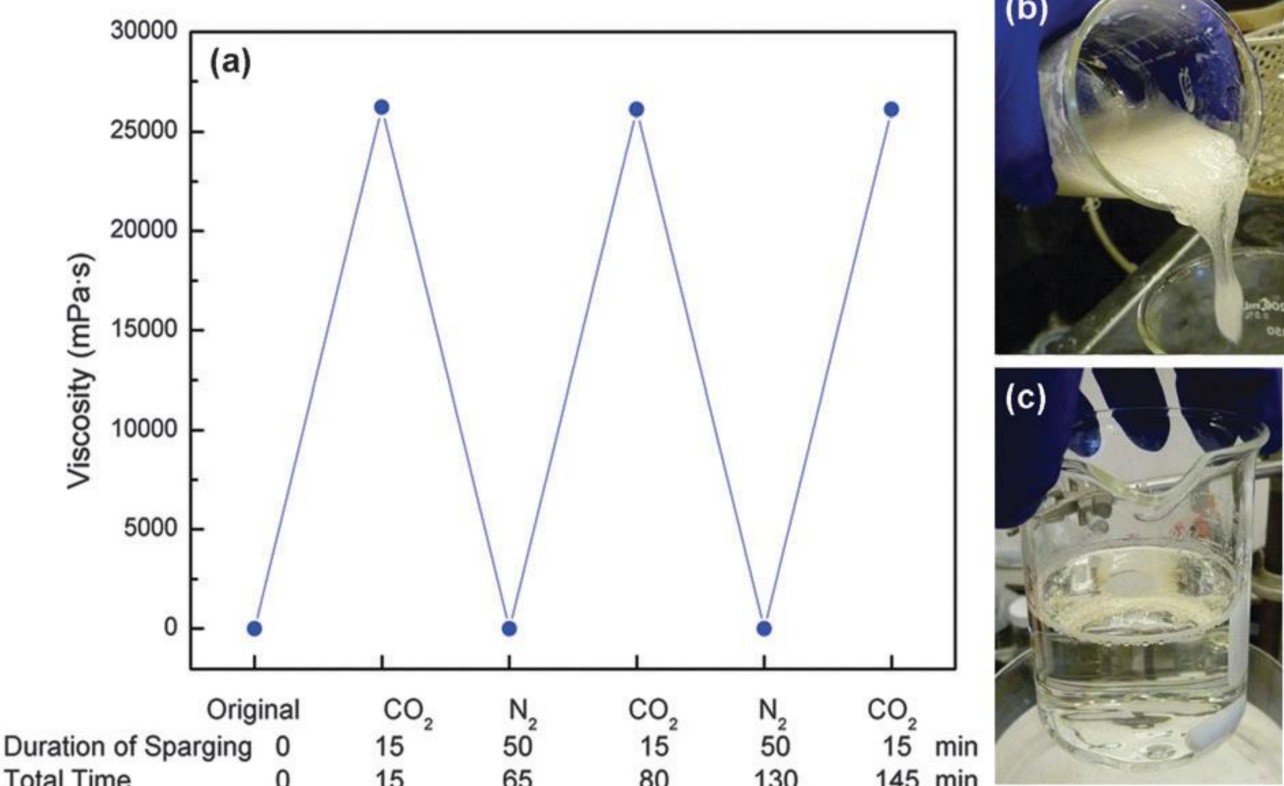

**Figure 11.** The aqueous solution of C18SNa (200 mM) and DMAE (200 mM) was transitioned from low and high zero-shear viscosity phases by switching $CO_2$ (1 bar) and $N_2$ (1 bar) treatments for three series at 60 °C (**a**); The water solution was produced by mixing 100 mL distilled water with $C_{18}$SNa and manually agitating it at 60 °C for several minutes. It was viscous, with a viscosity of 1.1 mPas. After adding the DMAE, the viscosity was 1.2 mPas. After 15 minutes of sparging $CO_2$ at 60 °C, the solution formed a gel with a viscosity of 26200 mPa·s. (**b**); the viscosity was restored to its previous value of 1.2 mPa·s after sparging $N_2$ for 50 minutes at 60 °C (**c**). "Reproduced with permission from Ref. [113]; 2022, Royal Society of Chemistry".

Figure 12 shows the steady shear viscoelastic properties of three EADP aqueous systems due to shear rate. As can be observed, the solutions have a characteristic viscosity independent of the shear rate at low shear rates following $CO_2$ bubbling. At low shear rates, it behaves Newtonian. The viscosity of the EADP solution exhibits a shear-thinning phenomenon once the shear rate is more significant than $0.1 \text{ s}^{-1}$, which can be interpreted as an indication of wormlike micelles that experience a phase transformation the aligning of elongated micelles at a high shear rate. The viscosities of the EADP solution improved throughout the measuring range after 50 mM NaSal was added, which was not the case with the EADP-only system. At low shear rates, the viscosities of the EADP solution increased by order of magnitude when 50 mM Maleic acid (MA) was introduced. At lower shear rates, the viscosity remains essentially unaltered and is nearly equal to the zero-shear viscosity, resulting in Newtonian behavior. The zero shear viscosities of three components execute the sequence MA > NaSal > EADP when $CO_2$ is present. The viscosity of the solution is low, regardless of the shear rate before the $CO_2$ bubbles, resulting in Newtonian fluid behavior.

Shaikh et al. [103] performed different rheological experiments to see how different high shear rates (170 and $510 \text{ s}^{-1}$) affected the apparent viscosity of the created fracturing fluid (SDS-TMTAD-$CO_2$) at 25 °C. The experiment was subdivided into four 600 s time segments. At a shear rate of $170 \text{ s}^{-1}$, the viscosity rose to 163 mPa·s in the first time segment (0–600 s). While at a shear rate of $510 \text{ s}^{-1}$, the viscosity dropped to 105 mPa·s

during the second period (600–1200 s), indicating acceptable viscous behavior against the high shear rate. During the third segment (1200–1800 s), the shear rate was reduced from 510 to 170 $s^{-1}$ and remained constant. The viscosity against increased to 168 mPa·s in less than a second after the shear rate was reduced, as shown in Figure 13. The shear rate was raised from 170 to 510 $s^{-1}$ in the last time segment (1800–2400 s), whereas the viscosity declined significantly and remained constant at 107 mPa·s during the whole interval. According to the phenomena, the fluid recovers great viscosity against high shear rates. The interfacial tension ($2.3 \times 10^{-2}$ mN m$^{-1}$) has good conditions during the flow-back time, clay swelling, and residual oil saturation, respectively.

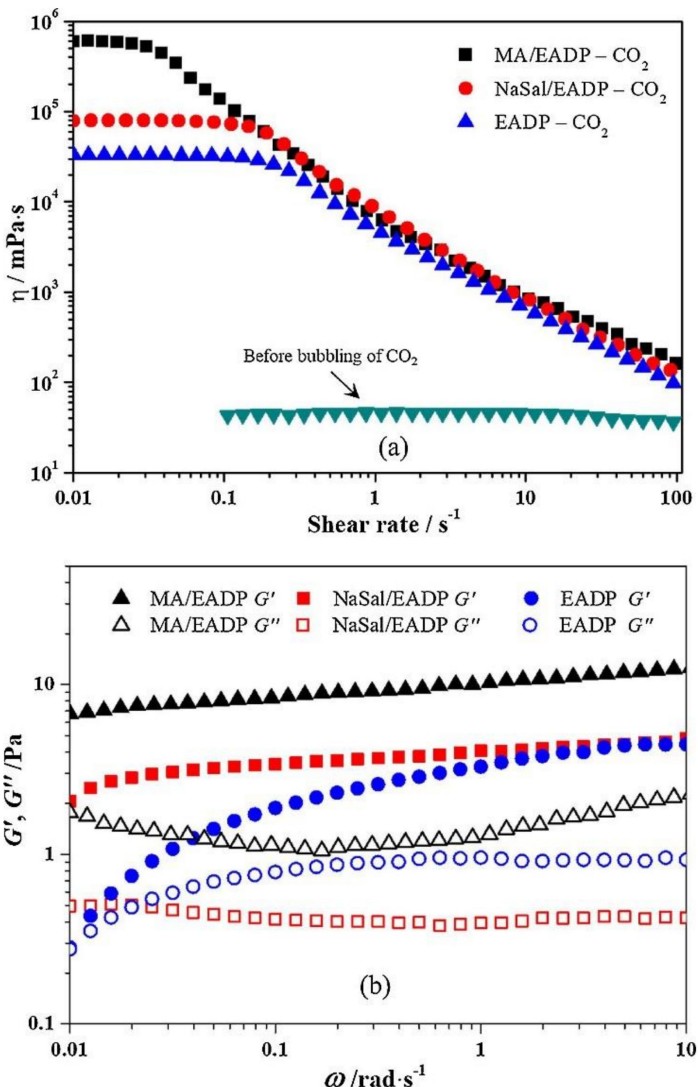

**Figure 12.** Shear viscosities in steady-state for 100 mM EADP aqueous systems with various additives (**a**); For EADP solutions with various additives, storage and loss modulus versus frequency (**b**) [115].

## 3.5. Effect of Salts

The efficiency of fracturing fluids is frequently affected by high salinity in formation water. The viscosity of erucamidopropyl dimethylamine (EA) was measured at a shear rate of 20 $s^{-1}$ to investigate the effect of salt concentration on $CO_2$-SVFFs [32]. The viscosity of the solution had a minimal impact when the salt quantity was low. When salt concentrations increased, the viscosity of the solution decreased somewhat, indicating that the network structure of VES fluids was slightly disrupted, as shown in Figure 14a. With divalent ions, the viscosity of $CO_2$-SVFFs decreases to a lesser extent compare to monovalent ions as in Figure 14b. The primary explanation for this could be that EA will shift from nonionic to

cationic surfactant in $CO_2$ and water, and salt input might affect the cationic head group by screening the electrostatic connection between the cation charged surfactant head groups via the salt electrolyte; as a result, the system's viscosity is reduced slightly [116]. As a result, this Viscoelastic fluid can tolerate high salinity.

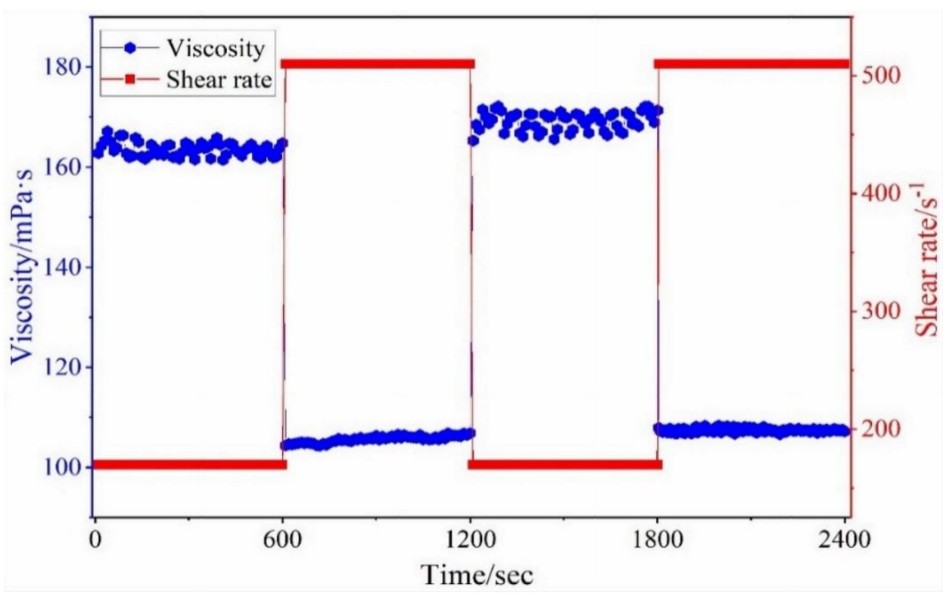

**Figure 13.** Shear resistance test at 25 °C [103].

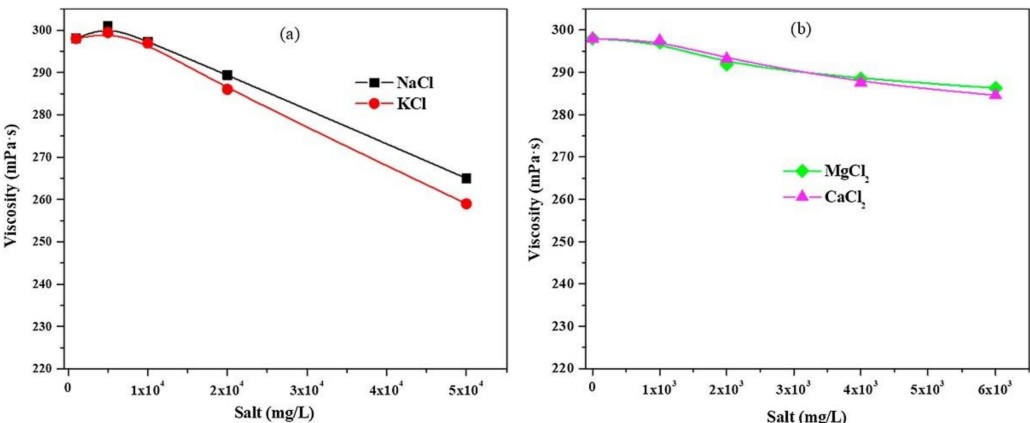

**Figure 14.** Shear viscosity of a 3% EA solution with inorganic salts: (**a**) KCl and NaCl are monovalent salts; (**b**) $MgCl_2$ and $CaCl_2$ are divalent salts [32].

### 3.6. Effect of Temperature and Pressure

The temperature has a more significant impact on surfactant molecular thermodynamics. As the temperature rises, the interactions between the head groups of surfactant molecules at the surface weaken [117]. Figure 15 shows the effect of shear time, shear rates, pressure, and temperature on the viscosity of $CO_2$-SVFFs. The rheological properties of $CO_2$-SVFFs are also significantly influenced by pressure. Without $CO_2$ pressure, a 2% erucamidopropyl dimethylamine (EA) aqueous solution behaves such as a water-like Newtonian fluid. When 800 psi of $CO_2$ was added to the system, the fluid's viscosity at a shear rate of $20\,s^{-1}$ increased by 2-fold compared to the $CO_2$-free system. The different shear rates affected the fluid's viscosity [32]. The VES fracturing fluid's good rheological performance under supercritical $CO_2$ circumstances shows much potential for fracturing applications.

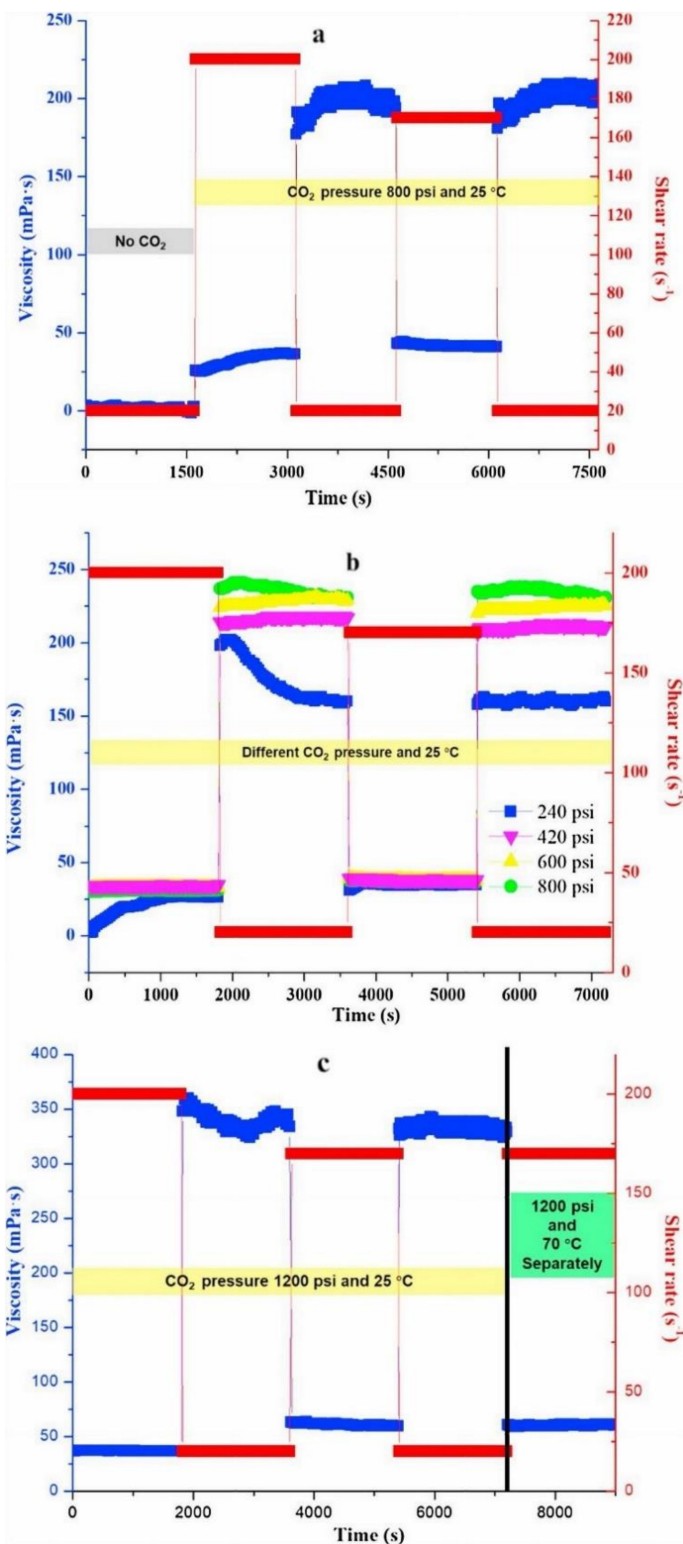

**Figure 15.** Viscosity plots to time for various shear rates and $CO_2$ pressures and temperatures. (**a**) At 800 pressure and 25 °C, the viscosity of a 2 percent in an aqueous solution of EA with no $CO_2$ was low but improved dramatically after bubbling $CO_2$. (**b**) The viscosity of 3% EA at various $CO_2$ pressures and temperatures of 25 °C. (**c**) The viscosity of a 3% EA aqueous solution at 25 °C and 70 °C under $CO_2$ pressure of 1200 psi [32].

### 3.7. Viscoelastic Behavior

The fluid's viscosity and elasticity play an essential role in suspending the proppants particles in VES fluids [13]. Due to an apparent viscoelastic response (i.e., the behavior is elastic (storage modulus G′ > loss modulus G″) at high frequencies, the dynamic rheology of VES aqueous solution in the presence of $CO_2$ results in the formation of wormlike micelles [37]. Researchers attempted to endow $CO_2$-reversible responsiveness to the CTAB-NaSal worm (TEA) [117]. The CTAB-NaSal worm was fragmented in the absence of $CO_2$. It displayed a water-like solution with the appropriate TEA concentration before recovering to highly viscoelastic fluid as bubbling $CO_2$ due to the shift between spherical micelles and WLMs. Figure 16 demonstrates that bubbling $CO_2$ generates a mainly viscoelastic response at high shear frequencies, with G′ exceeding G″.

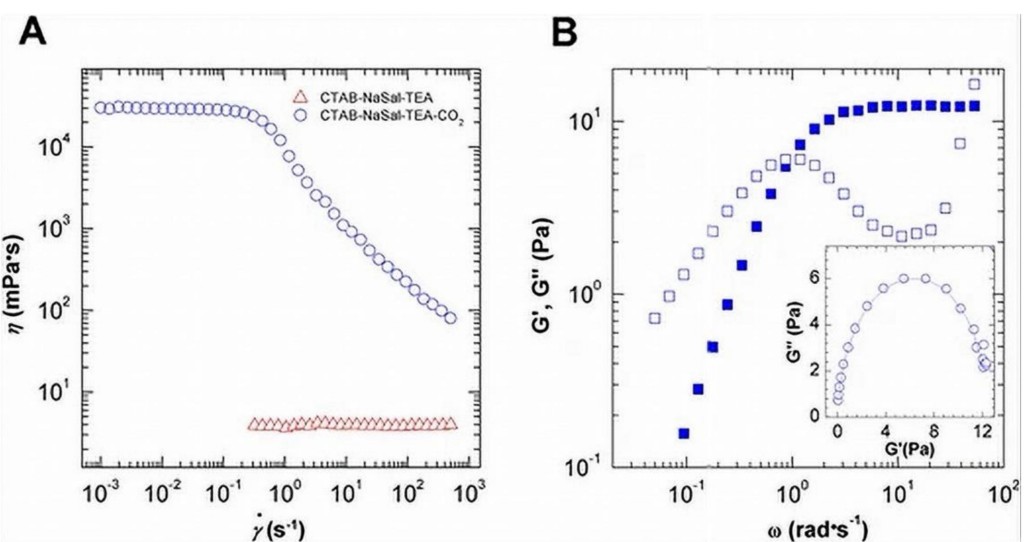

**Figure 16.** (**A**) Static rheology and (**B**) dynamic rheology of 50 mM CTAB-NaSal-TEA before and after bubbling $CO_2$ at 25 °C (with a molar ratio of 1:1:5) [117].

In a study, different experiments were conducted to explore the viscoelastic behavior of solutions of *N*-butyldiethanolamine–sodium oleate (BDEA–NaOA) and *N,N*-diethyl butylamine–sodium oleate (DEBA–NaOA) with $CO_2$ effect [95]. The pH value reached 8.45, the viscosity remained constant with a rising shear rate, and the solution acted such as water. This type of behavior is characteristic of Newtonian fluids. When the pH value is between 8.65 and 8.45, and the shear rate rises, all samples exhibit Newtonian behavior at first, then shear-thinning behavior at a higher shear rate. The shear-thinning behavior of the wormlike micelles was characteristic of wormlike micelles, showing that wormlike micelles linked with each other in the solution.

### 3.8. Effect of Gel Structure Breaking and Proppant Suspension and Carrying Capacity

The WLMs in the VES fracturing fluid disintegrate into spherical micelles on interaction with the produced hydrocarbons during flow back, leading to a low viscosity fluid that is easier to remove from the pore space propped fracture [86]. The VES fracturing fluid does not require chemical breakers, and gel breaking is performed by adjusting the fluid-$CO_2$ interaction [32]. For an excellent Flow back process, the breaking fluid viscosity must be according to the standard, i.e., less than 5 mPa·s [103], as shown the Figure 17. The ability of a fracturing fluid to suspend proppants is a fundamental attribute that impacts the quality of fracturing construction. The viscoelastic qualities of the VES fluid are ideal for proppant transport. $CO_2$-SVFFs can perform two functions: high viscosity for fracking and proppant transportation and low viscosity for fracking fluid recovery [32].

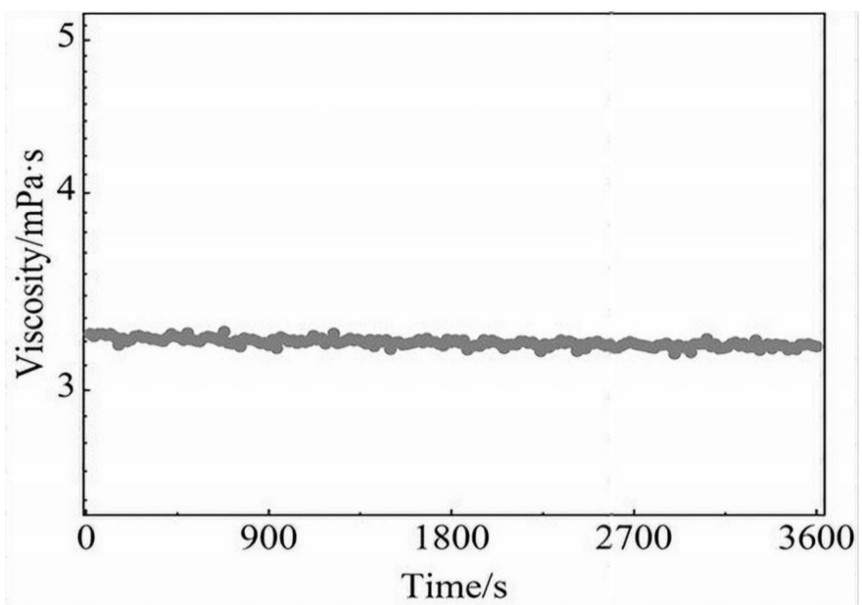

**Figure 17.** Viscosity measurement of gel breaking fluid using kerosene, 25 °C [103].

*3.9. Filtration and Permeability Damage Evaluation*

Filtration assessment is the most crucial parameter and remains one of the most critical qualities for evaluating reservoir fracturing strategies. If fluid leak-off behavior decreases bottom hole pressure and unequal proppant dispersion, the fracturing work will perform poorly. Different studies analyzed the literature's filtration of $CO_2$-SVFFs, and the results were outstanding compared to conventional fracturing fluids [103]. It has already been mentioned that if the fracturing fluid is not formation friendly, it will cause problems with fracture conductivity and damage the formation by lowering the formation's permeability [86]. Therefore, the fracturing fluid with low formation damage characteristics is essential in the fracturing technique. Different researchers calculated the formation damage of $CO_2$-SVFFs using the following Equation (1) [32,41,118].

$$\eta_{d=} \frac{k_1 - k_2}{k_1} \times 100 \tag{1}$$

where $\eta_d$ is the formation damage ratio in percentage, while $k_1$ and $k_2$ are the permeability of the core before and after fluid injection, respectively.

**4. Advantages and Limitations**

$CO_2$-SVFFs are more effective because of their higher retained proppant pack permeability and controllable rheological behavior. When fracturing with $CO_2$-SVFF, viscosity plays a major role in providing sufficient fracture width to ensure proppant entrance into the fracture, carrying the proppant from the wellbore to the fracture tip, generating a desired net pressure to control height growth, and providing fluid loss control. The $CO_2$-SVFF is based on the WLMs formation, achieving zero residues, and obtaining highly conductive stimulation treatments with no polymer damage. $CO_2$-SVFFs reduce the cost of cleaning, which was found in the water and polymer-based fracturing fluid. Abundant $CO_2$ can potentially be green which can be used at low cost with a minor amount of surfactant and its counter ion additives. The physical association and entanglement of worm-like micelles present excellent viscoelastic properties of proppant transport, which is better than guar-based fluids.

On the other hand, the fluid system is the most widely used imbibition drainage and displacement system for low permeability oil reservoirs. It could effectively change the wettability of formation and reduce the oil-water interfacial tension, which improves oil

recovery. Coupling the benefits of a VES fluid with $CO_2$, the emulsified system will further enhance cleanup in a depleted reservoir, extend the application to water-sensitive formations, and maintain reservoir gas saturation to prevent potential water blocks. $CO_2$-sensitive clean fracturing fluid systems exhibit lower friction pressure, superior proppant transport, higher fracture conductivity, and longer effective fracture half-length than conventional systems by eliminating polymer. Both $CO_2$ and surfactant can improve low permeability oil recovery by diffusion and imbibition drainage displacement, respectively, which is economically and technically viable. $CO_2$ with surfactants can dramatically improve $CO_2$ utilization, surfactant cost reduction, and oil recovery. Given the circumstances, the most recent advancement in fracturing fluids is an ecologically benign and non-polluting $CO_2$-based clear fracturing fluid.

However, the $CO_2$-SVFFs technique requires high pump energy compared to the $CO_2$ fracturing method. Since $CO_2$ is very mobile, this approach has issues with viscous fingering and gravity overriding due to the limited capacity to control $CO_2$ mobility. When high-pressure $CO_2$ is required for VESs, there are issues with handling and safety. When the pressure is released, the $CO_2$ condenses into dry ice plugs.

## 5. Future Recommendations

Different researchers used various strategies to improve fluid characteristics, and positive results were observed. More research is needed at high temperatures and pressures for $CO_2$-SVFFs. The academia and industry must work together to produce VES fluids that respond differently to external stimuli to reduce costs while improving $CO_2$-SVFF performance in extended reservoir conditions. Furthermore, much experimental research of VES fluids for fracturing concentrated on common aspects such as how temperature impacts the $CO_2$-SVFFs properties. The studies should be broad to different reservoir conditions. At high pH, the viscosity of $CO_2$ decreases, so better counter ions should be added to tackle this problem. There has been very little research on $CO_2$-SVFF with divalent ions. Further research should include the wettability alteration by $CO_2$-SVFFs.

Conventional viscoelastic surfactant fluids, which are utilized in large quantities, the viscoelasticity property is reduced when exposed to high temperatures. The Gemini $CO_2$-sensitive VES can be employed in small amounts for better results. The rheological characteristics of $CO_2$-SVFFs for high-temperature reservoirs and VES systems are required by various hydrophobic chains linked by a spacer group. Nanoparticles should be added to $CO_2$-SVFFs to further improve their rheological properties. High-temperature reservoirs tri-cationic surfactants with the required amount of organic salt exhibit excellent rheology and accumulation properties. So further research is needed to use tri-cationic surfactants in $CO_2$-SVFFs. The hydraulic fracturing technique is of great importance in developing of shale oil and gas recovery. Further studies of $CO_2$-SVFFs in unconventional reservoirs, especially shale, are needed. There is a lack of studies of $CO_2$-SVFFs with temperature ($\geq 120$ °C) and shear rate ($\geq 500$ s$^{-1}$). So further studies should also be focused on high temperature and shear rate.

## 6. Conclusions

In contrast with conventional surfactants, $CO_2$-SVFFs are switchable between high viscosity for fracture and proppant transport and low viscosity for easy removal of fluid after inducing fracture. $CO_2$-SVFFs reduce the amount of water needed to make VES fluids by combining a small quantity of water with a large volume of $CO_2$. The good viscoelastic properties of $CO_2$-SVFFs are exhibited essentially for proppant transport. The inclusion of inorganic salt ions shows that the fluids have salinity tolerance with a moderate reduction in zero-shear viscosity. $CO_2$ is a relatively inexpensive trigger that is easy to remove and does not accumulate or contaminate waste streams.

The $CO_2$-SVFF has high shear tolerance, thermal stability, moderate salinity tolerance, and reduced core damage features. For gel breaking performance at 90 °C, the mixing of kerosene/standard brine with different proportions in $CO_2$-SVFFs results in a more

remarkable ability to lower the viscosity as per the standard of fracturing fluids ($\leq 5$ mPa·s for nearly 2 h). Increase in pressure of $CO_2$-SVFF, the rheological coefficient ($k'$) and effective viscosity ($\eta e$) increased, but the rheological index ($n'$) decreased. For higher temperatures ($\geq 120$ °C) and strong salinity formations (exceeding 2% $CaCl_2$, 3% NaCl, and $MgCl_2$), $CO_2$-SVFF with a low concentration of nanoparticles (NPs-$CO_2$-SVFF) is recommended to apply. Many studies focus on the laboratory scale; thus, researchers should evaluate the assessment of $CO_2$-SVFF for fracturing in numerous oilfields.

**Author Contributions:** Conceptualization, L.Z.; methodology, validation, resources, data curation, and writing-original draft writing preparation, A.B., H.W., N.K. and Z.K.; writing-review and editing, visualization, and supervision, A.S., R.S. and L.Z.; project administration, L.Z. All authors have read and agreed to the published version of the manuscript.

**Funding:** This research was supported by the "General project of Shandong Natural Science Foundation (ZR2020ME090), the Fundamental Research Funds for the Central Universities (No. 17CX06006), the graduate innovation funding project from China University of Petroleum (East China) (YCX2017022) the National Oil and Gas Major Projects (No. 2016ZX05056004-003), and the Changjiang Scholars and Innovative Research (IRT 1294 and 1086/14R58).

**Institutional Review Board Statement:** Not applicable.

**Informed Consent Statement:** Not applicable.

**Data Availability Statement:** Not applicable.

**Conflicts of Interest:** None of the authors of this paper has a financial or personal relationship with other people or organizations that could inappropriately influence or bias the content of the paper.

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
