# Peer review of "Development of CO2-Sensitive Viscoelastic Fracturing Fluid for Low Permeability Reservoirs: A Review"

_processes, doi:10.3390/pr10050885_

Round 1

Reviewer 1 Report

This is a nice piece of review work. However, I would expect some more details on the effect of additives and other parameters should be included.

The second concern is on the quality of the figures/images. The image quality should substantially be improved. They seem to have been copied from literature.

Author Response

Respected Reviewer

On behalf of my coauthors, I would like to thank you for the opportunity to revise and resubmit our manuscript processes-1622171, entitled “Development of CO2-Sensitive Viscoelastic Fracturing Fluid for Low Permeability Reservoirs: A Review” for publication in “Processes”

We found the editor's comments to help revise the manuscript and have carefully considered and responded to each suggestion. We have made all the changes as you were suggested to us.

Thank you again for your consideration of our revised manuscript.

With my best regards

Yours sincerely

Liang Zhang

[email protected]

Reviewer 2 Report

The manuscript “Development of CO2-Sensitive Viscoelastic Fracturing Fluid for Unconventional Reservoirs: A Review” reviews Viscoelastic surfactants and CO2-sensitive viscoelastic fracturing development and applications, and presents a comparison of benefits, limitations, and drawbacks to conventional fracturing fluids. The paper is well written and organized but needs improvement. The following are my recommendations:

  1. Line73: This paragraph repeats the sentence at Line 60. Authors should decide which one to keep or paraphrase it.
  2. Figures: All the figures on the manuscript need improvement. Their quality is very low and sometimes hard to read. Also, as in figures 6,9,12,15,17, the authors changed the graphs' size, resulting in modified scales that might be deceiving.
  3. Figure captions: Some of the figure captions are not below the figures; they should all be below the figures (Figure 16 and 18)
  4. Line 111: What does WLM stand for? Authors should provide the full wording of any abbreviation when it’s first used.
  5. Line 137: In the abstract (Line 16), the authors stated, “The primary aspect of the hydraulic fracturing plan is to develop clean and less expensive fracturing fluid” but in this line they are disclosing that VES has high cost and environmental problems. These issues were not revisited in the text. Authors should add a paragraph or a section explaining why VES fracturing fluids (with or without CO2 responsive) might be cheaper and cleaner.
  6. Line 175: The statement “vaporizes components of crude oil” is not clear. Can the authors elaborate on what they mean by vaporizing oil components?
  7. Line 176: In this paragraph, the authors summarize the studies and some field tests on the improvements of CO2 on VES fluids. However, the cited studies are rather old, and most are applied to conventional reservoirs. So, the technology, reservoir response, processes are different. In hydraulic fracturing of unconventional reservoirs, the need for fracturing fluid is relatively high. Can the authors also include information about the application of CO2-sensitive VES on unconventional reservoirs?
  8. Line 198: What does TMTAD stand for? Authors should provide the full wording of any abbreviation when it’s first used.
  9. Line 238: The authors start the paragraph by describing immiscible gas (CO2) flooding for pressure maintenance as an IOR method. Line 241 mentions the MMP of CO2, which is an essential factor for miscible gas injection as an EOR method. These two methods are entirely different and applied for other reasons at different times of the lifecycle of a reservoir. The authors should restructure this paragraph not to cause misunderstanding.
  10. Line 245: Authors state that CO2 is non-corrosive. This is true for dry CO2, but when CO2 reacts with water (e.g., formation brine), carbonic acid will be formed, becoming corrosive. This is one of the main issues of CO2 EOR. Therefore, the authors should change their statement here.
  11. Line 246: Reference 71- Yuan, B.; Su, Y.; Ghanbarnezhad, R.; Rui, Z.; Wang, W.; Shang, Y. Journal of Natural Gas Science and Engineering A new analytical multi-linear solution for gas flow toward fractured horizontal wells with different fracture intensity. J. Nat. Gas Sci. Eng. 2015, 23, 227–238, doi:10.1016/j.jngse.2015.01.045.- does not have a statement used in the manuscript. Can the authors add the correct reference here if there was a typo?
  12. Line 470: When authors mention the drawbacks, they state the need for high pump energy. This is because of the need to reach very high pressures and the need for high volumes to fracture an unconventional reservoir, which is sometimes even hard with conventional hydraulic fracturing methods. Can the authors elaborate on this aspect to provide a better understanding to the readers?

Author Response

Respected Reviewer,

On behalf of my coauthors, I would like to thank you for the opportunity to revise and resubmit our manuscript processes-1622171, entitled “Development of CO2-Sensitive Viscoelastic Fracturing Fluid for Low Permeability Reservoirs: A Review” for publication in “Processes”

We found the editor's comments to help revise the manuscript and have carefully considered and responded to each suggestion. We have made all the changes as you were suggested to us.

Thank you again for your consideration of our revised manuscript.

With my best regards

Yours sincerely

Liang Zhang

[email protected]

Reviewer 3 Report

Title: Development of CO2-Sensitive Viscoelastic Fracturing Fluid for Unconventional Reservoirs: A Review (processes-1622171)

In this article, Allah Baksh et al. tried to review the application of CO2-sensitive viscoelastic fracturing fluids (CO2-SVFF) in the hydraulic fracturing operation. The effect of different operating parameters (pH, conductivity, temperature, and rheological behaviors of injected fluids) has also been investigated.

I know that preparing a manuscript needs a lot of time and effort, but there are some basic issues that convinced me to suggest rejection for this review article. Some of my concerns are below:

  • Writing a good review paper needs the authors to have 10-15 articles on the considered topic. The authors of this work only published 2-3 papers in this field which is not enough for organizing a review paper.
  • Interaction between CO2-Sensitive Viscoelastic Fracturing Fluid and reservoir rock has been missed to include. Moreover, little attention has been paid to explaining the CO2-SVFF and reservoir fluids in the high pressure and high pressure, and high salinity conditions.
  • The compatibility between CO2-SVFF and reservoir rock and fluids should be reviewed deeply.
  • It is not a good idea to cite so many references in a little text. You should break them into some extensive expressions and clearly state which reference is about which expression.
  • References in the text are organized badly. For example, see Page 1, Line 42.
  • Following the previous comment, numbered the references in the main manuscript continuously. For example, it seems [28, 29], [45], [54, 55], and so many others are missed.
  • Figures' quality is not good. Moreover, you should prepare some graphs/flowcharts to clearly explain your statement.
  • Add a new section devoted to the future and required steps in this field.
  • You reviewed a few articles published during the five recent years (2018-2022). I think you should mainly focus on the articles published in this duration.
  • References do not have the same style. Try to only use one standard style for the bibliography.
  • I really expected to see some explanations about the molecular-scale interaction between CO2-SVFF and reservoir rock and fluids. For example, mechanism of wettability alteration, possible surface reactions, and so on.

Author Response

(The authors gave the same response as above.)

Round 2

Reviewer 2 Report

Thank you for making the editions on your manuscript.

Author Response

20-April-2022

Dear reviewer,

Thank you very much for the valuable comments and the constructive suggestions on our manuscript entitled “Development of CO2-Sensitive Viscoelastic Fracturing Fluid for Low Permeability Reservoirs: A Review” (processes-1622171). We appreciate you and the reviewers for your precious time in reviewing our paper and providing valuable comments. Your valuable and insightful comments led to possible improvements in the current version. The authors have carefully considered the comments and tried their best to address every one of them. We hope the manuscript after careful revisions meet your high standards. The authors welcome further constructive comments if any.

Best regards,

Dr. Liang Zhang

School of Petroleum Engineering

China University of Petroleum (East China)

No 66 ChangJiangXi Road, Huangdao District

Qingdao 266580, P R China

Email: [email protected], [email protected]

Reviewer 3 Report

Reject.

Author Response

20-April-2022

Dear reviewer,

Thank you very much for the valuable comments and the constructive suggestions on our manuscript entitled “Development of CO2-Sensitive Viscoelastic Fracturing Fluid for Low Permeability Reservoirs: A Review” (processes-1622171). We appreciate you and the reviewers for your precious time in reviewing our paper and providing valuable comments. Your valuable and insightful comments led to possible improvements in the current version. The authors have carefully considered the comments and tried their best to address every one of them. We hope the manuscript after careful revisions meet your high standards. The authors welcome further constructive comments if any.

Best regards,

Dr. Liang Zhang

School of Petroleum Engineering

China University of Petroleum (East China)

No 66 ChangJiangXi Road, Huangdao District

Qingdao 266580, P R China

Email: [email protected], [email protected]

Tel: +86 1505325974